# Cas12a-assisted precise targeted cloning using in vivo Cre-*lox* recombination

Behnam Enghiad[1,2,4], Chunshuai Huang[1,4], Fang Guo[2], Guangde Jiang[1], Bin Wang [1], S. Kasra Tabatabaei[1], Teresa A. Martin[2] & Huimin Zhao [1,2,3✉]

Direct cloning represents the most efficient strategy to access the vast number of uncharacterized natural product biosynthetic gene clusters (BGCs) for the discovery of novel bioactive compounds. However, due to their large size, repetitive nature, or high GC-content, large-scale cloning of these BGCs remains an overwhelming challenge. Here, we report a scalable direct cloning method named Cas12a-assisted precise targeted cloning using in vivo Cre-*lox* recombination (CAPTURE) which consists of Cas12a digestion, a DNA assembly approach termed T4 polymerase exo + fill-in DNA assembly, and Cre-*lox* in vivo DNA circularization. We apply this method to clone 47 BGCs ranging from 10 to 113 kb from both Actinomycetes and Bacilli with ~100% efficiency. Heterologous expression of cloned BGCs leads to the discovery of 15 previously uncharacterized natural products including six cyclic head-to-tail heterodimers with a unique 5/6/6/6/5 pentacyclic carbon skeleton, designated as bipentaromycins A–F. Four of the bipentaromycins show strong antimicrobial activity to both Gram-positive and Gram-negative bacteria such as methicillin-resistant *Staphylococcus aureus*, vancomycinresistant *Enterococcus faecium*, and bioweapon *Bacillus anthracis*. Due to its robustness and efficiency, our direct cloning method coupled with heterologous expression provides an effective strategy for large-scale discovery of novel natural products.

[1] Carl R. Woese Institute for Genomic Biology, University of Illinois at Urbana-Champaign, Urbana, IL, USA. [2] Department of Chemical and Biomolecular Engineering, University of Illinois at Urbana-Champaign, Urbana, IL, USA. [3] Departments of Chemistry, Biochemistry, and Bioengineering, University of Illinois at Urbana-Champaign, Urbana, IL, USA. [4] These authors contributed equally: Behnam Enghiad, Chunshuai Huang. ✉email: zhao5@illinois.edu

Natural products or small molecules produced by living organisms have played an important role in human medicine and agriculture for centuries. Recent analysis of all FDA approved small molecule drugs between 1981 and 2019 has revealed that ~34% of these compounds are either natural products or natural products derivatives and a significant number of these compounds are produced by microorganisms[1]. With recent advances in genome sequencing and bioinformatics tools, microbial natural product biosynthetic gene clusters (BGCs) are uncovered at an unprecedented rate. However, only a relatively small fraction of these BGCs have been associated with known products[2–4]. As a result, microorganisms potentially possess a huge reservoir of currently uncharacterized novel natural products[5,6].

To date, a wide variety of approaches have been developed to investigate these uncharacterized BGCs[6,7], which can be grouped into native host based[8,9] and heterologous host based[10,11] strategies. Due to lack of developed genetic tools, the native host based strategies are not generally applicable to most microorganisms. In contrast, since BGCs can be cloned from purified genomic DNA, the heterologous host based strategies can be applied to BGCs from all microorganisms including those yet to be cultivated in the laboratory. Traditionally, microbial BGCs have been cloned by construction of genomic DNA libraries and subsequent screening to find the correct clone[12,13]. Although this random cloning strategy has successfully been used for cloning microbial BGCs, it is not suitable for large-scale discovery of novel natural products due to its untargeted nature and extensive screening requirement. Recently, with the availability of public genome sequences, a variety of methods for direct cloning of microbial BGCs have been developed[14]. In vivo homologous recombination strategies including transformation-associated recombination (TAR) in *Saccharomyces cerevisiae*[10,15] and linear-linear homologous recombination (LLHR) in *Escherichia coli*[16,17] take advantage of the high recombination efficiency of the cloning host to clone genomic fragments into plasmids. Unfortunately, many microbial BGCs have repetitive DNA sequences that can negatively interfere with successful recombination events and acquisition of correct clones using these strategies[15]. Alternatively, microbial BGCs can also be cloned using in vitro strategies. Cas9-assisted targeting of chromosomal segments (CATCH) method[18] uses a combination of in vitro Cas9 digestion of genomic DNA in agarose gel-plugs and isothermal Gibson assembly[19] for cloning genomic DNA fragments into linearized DNA vectors. Although this approach can potentially be used to clone BGCs with repetitive DNA sequences, due to the low efficiency of Gibson assembly in assembling DNA molecules with high GC-content[20], this approach is not suitable for cloning large BGCs (>50 kb) from high GC-content organisms including Actinobacteria, one of the most gifted producers of natural products[21].

Here, we report an inexpensive, rapid, robust, and highly efficient direct cloning method named Cas12a-assisted precise targeted cloning using in vivo Cre-*lox* recombination (CAPTURE), which allows direct cloning of large genomic fragments (up to 113 kb) including those with high GC-content and sequence repeats into functional vectors. We characterize the efficiency and robustness of CAPTURE by cloning 47 natural product BGCs from both Actinomycetes and Bacilli. We also demonstrate the utility of CAPTURE in large-scale discovery of natural products by direct cloning and heterologous expression of 43 uncharacterized microbial BGCs, which has led to the discovery of 15 previously uncharacterized natural products.

## Results

**Design of the CAPTURE method**. The workflow of CAPTURE is shown in Fig. 1a. In the first step, purified microbial genomic DNA is digested by Cas12a enzyme to release the BGC fragment. In the second step, two DNA receivers are amplified by polymerase chain reaction (PCR). The DNA receivers each carry a *lox*P site at their ends. To limit the possibility of creating false positive colonies, none of the DNA receivers carry both resistance marker and *E. coli* origin of replication. Other than these two essential elements, DNA receivers can also carry elements required for heterologous expression. This strategy would allow direct cloning of BGCs into vectors ready for heterologous expression. The DNA receivers are amplified from a set of universal receiver plasmids which can be designed for a specific heterologous host. Examples of universal receiver plasmids designed for *Streptomyces* and *Bacillus subtilis* heterologous hosts are shown in Supplementary Fig. 1. After PCR, the amplified DNA receivers are mixed with the digested genomic DNA in a T4 polymerase exo + fill-in DNA assembly reaction to join the three fragments and create a linear DNA product. In the third and final step, DNA assembly products are transformed into *E. coli* cells harboring a helper plasmid expressing Cre recombinase and phage lambda Red Gam protein for in vivo Cre-*lox* circularization. The resulting colonies are then checked for correct clones.

**Development of the CAPTURE method by creating an efficient in vivo DNA circularization system based on site-specific recombination**. The creation of a direct cloning method based on in vivo site-specific recombination was based on the fact that due to the creation of by-products such as DNA concatemers (e.g., Supplementary Fig. 2a), DNA circularization is the main limiting step in the assembly of large DNA molecules in vitro[22]. To potentially overcome this challenge, we sought to utilize a DNA circularization system in which instead of circular DNA, *E. coli* cells are transformed with linear DNA molecules which are able to circularize in vivo. By eliminating the formation of DNA by-products, this strategy can in turn increase the frequency of DNA circularization (Supplementary Fig. 2b).

In vivo circularization of linear DNA molecules can occur with the help of homologous recombination or site-specific recombinase enzymes[23]. Unlike homologous recombination which promotes recombination between all DNA sequences with high degrees of homology, site-specific recombination only occurs between defined DNA sequences which are recognized by the site-specific recombinase enzyme[24]. As a result, for DNA circularization, site-specific recombination offers an advantage over homologous recombination because of its ability in circularization of DNA molecules containing sequence repeats. Previously, a few systems based on site-specific recombinase proteins were developed for circularization of linear DNA molecules in *E. coli*[25,26]. However, these systems either have low circularization efficiency or lack transient expression of recombinase and require *RecBC* deficient strains of *E. coli*. Therefore, we decided to create a general transient *E. coli* circularization system by placing the required elements for DNA circularization on a plasmid (a.k.a. helper plasmid). This circularization helper plasmid (Fig. 1b) carries three essential elements: (1) a site-specific recombinase to carry out DNA circularization; (2) phage lambda Red *gam* gene for inhibition of linear DNA degradation by *E. coli* RecBCD complex[27]; (3) pSC101 temperature-sensitive origin of replication to enable curing of *E. coli* cells from the helper plasmid after circularization[28]. We evaluated two frequently used site-specific recombinases, Flp and Cre, by using two test plasmids carrying either two FRT or *lox*P sites flanking a *lacZ* gene (Supplementary Figs. 3 and 4a). In the event of successful recombination, *lacZ* gene would be removed and colonies would show white color in blue-white screening. Thus the ratio of white colonies to the total number of

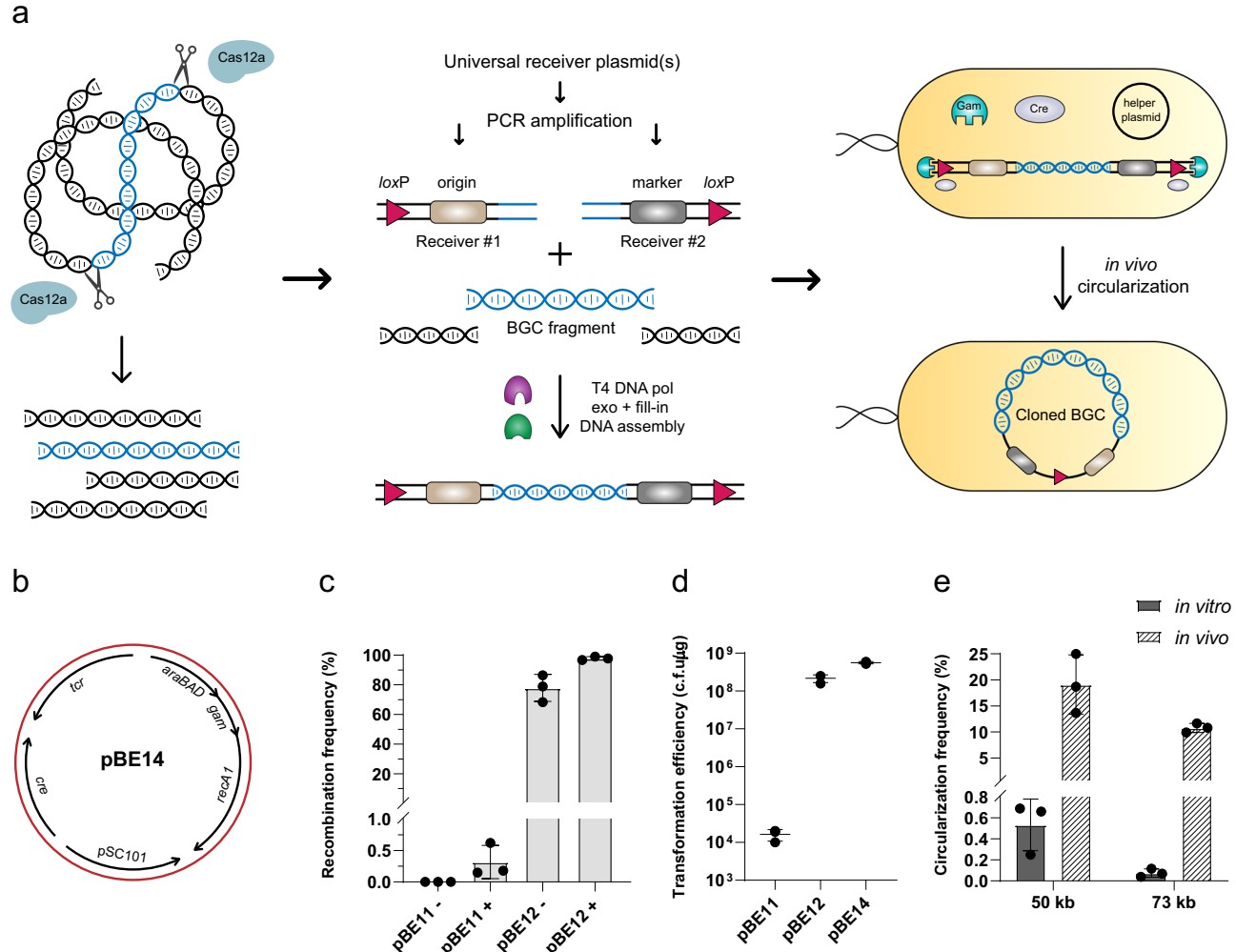

**Fig. 1 Development of the CAPTURE method. a** Overview of the workflow. In the first step, purified genomic DNA is digested by Cas12a enzyme to release the target BGC fragment. In the second step, digestion products are mixed with two DNA receivers containing *lox*P sites at their ends. The target BGC fragment and DNA receivers are assembled together using T4 DNA polymerase exo + fill-in DNA assembly. In the final step, the assembly mixture is transformed into *E. coli* cells harboring a circularization helper plasmid. The linear DNA is able to circularize in vivo by Cre-*lox* recombination. **b** DNA map of helper plasmid pBE14. *tcr*: tetracycline resistance marker; *araBAD*: L-arabinose inducible promoter and its regulator; *gam*: phage lambda Red gam gene; pSC101: temperature-sensitive origin of replication; *recA1*: mutated *E. coli* recA gene to increase transformation efficiency. **c** Comparison of recombination frequency between Flp (pBE11) and Cre (pBE12) helper plasmids. -: without L-arabinose induction, +: with L-arabinose induction. Recombination frequencies were calculated based on the ratio of white colonies to the total number of acquired colonies. **d** Linear DNA transformation efficiency for *E. coli* cells harboring pBE11 (Flp), pBE12 (Cre), pBE14 (Cre and recA1) helper plasmids. Both pBE12 and pBE14 *E. coli* cells exhibited transformation efficiencies similar to circular DNA. **e** Comparison of in vitro versus in vivo circularization for two large (50 kb, 73 kb) linear DNA molecules. In vivo circularization showed ~33-fold and 150-fold higher frequency than in vitro circularization for 50 kb and 73 kb molecules, respectively. Circularization frequencies were calculated based on the number of colonies acquired for each circularization experiment in comparison to the number of colonies acquired after transformation of the original circular DNA (see Methods for full description). Each experiment was performed in three biological replicates and data are presented as mean values ± standard deviation (SD). Source data are provided as a Source Data file.

acquired colonies can be used to calculate recombination frequency. As shown in Fig. 1c, when Flp recombinase expression was fully induced, we observed only ~0.3% recombination frequency for the test plasmid. Contrarily, even leaky expression of Cre recombinase from L-arabinose inducible promoter was enough to promote recombination at ~78% frequency and when the expression was fully induced, Cre recombinase was able to carry out recombination at close to 100% frequency. We next assessed the efficiency of Flp and Cre helper plasmids in circularization of linear DNA molecules. This time, the test plasmids were amplified by PCR and the linear PCR products (Supplementary Fig. 4b) were transformed into *E. coli* cells harboring Flp (pBE11) or Cre (pBE12 and pBE14) helper plasmids. As shown in Fig. 1d, Cre helper plasmids were able

to exhibit $10^8$–$10^9$ c.f.u/μg transformation efficiency for a 3 kb linear DNA molecule, which is essentially the same efficiency as transforming a circular DNA molecule with the same size (Supplementary Fig. 5).

Following creation of an efficient in vivo DNA circularization system in *E. coli*, we sought to directly compare the efficiency of DNA circularization in vitro and in vivo for large DNA molecules. For this purpose, two plasmid DNA molecules with sizes of 50 and 73 kb each carrying two *lox*P sites, were constructed (Supplementary Fig. 6). These plasmids were then linearized by restriction enzymes. For in vitro DNA circularization, the linearized DNA was ligated to itself using T4 DNA ligase and transformed into *E. coli* cells. For in vivo DNA circularization, the linearized DNA was transformed directly into *E. coli*

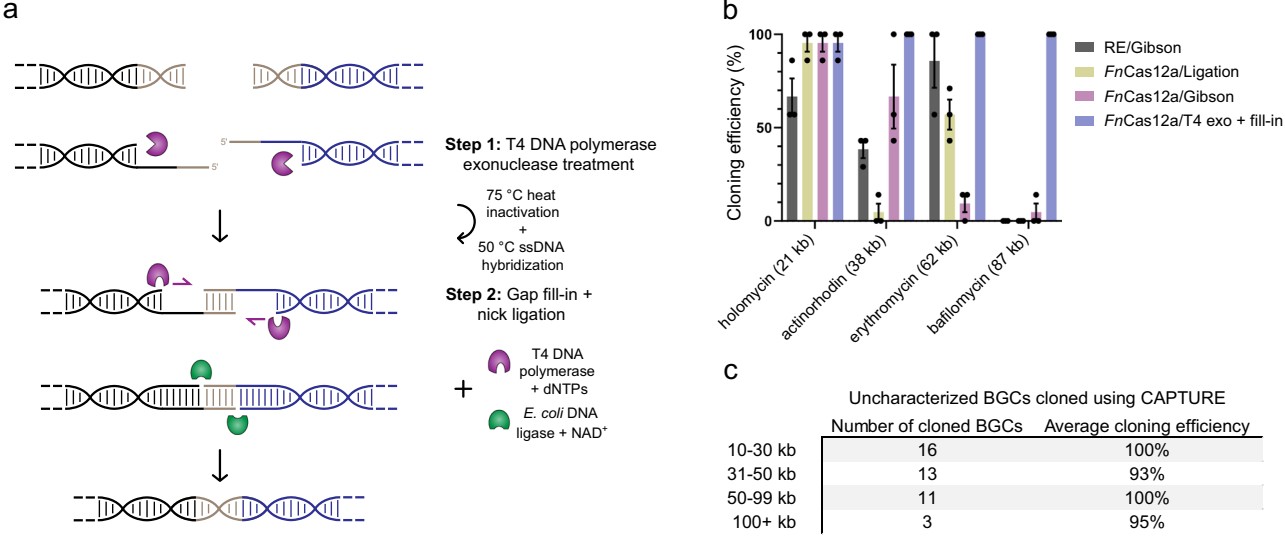

**Fig. 2 Characterization of various genomic DNA digestion/DNA assembly combinations in the CAPTURE method. a** Schematics of T4 DNA polymerase exo + fill-in DNA assembly. In step 1, DNA molecules ends are chewed back by T4 DNA polymerase to create ssDNA overhangs. The reaction mixture's temperature is increased to 75 °C to inactivate T4 DNA polymerase and potentially remove ssDNA secondary structures. Temperature is then decreased to 50 °C to allow for ssDNA overhang hybridization. In step 2, by addition of fresh T4 DNA polymerase, and dNTPs, DNA gaps in the hybridized DNA molecule are filled. *E. coli* DNA ligase is then used to ligate the nicks and produce the final assembly product. **b** Comparison of different digestion/DNA assembly combinations in cloning four high GC-content BGCs from Actinomycetes. The *Fn*Cas12a/T4 exo + fill-in strategy showed ~100% cloning efficiency for all four target BGCs. RE: restriction enzymes. For each cloning experiment, at least seven colonies were selected and the purified plasmids from each colony were analyzed by restriction digestion. The cloning efficiencies were calculated as the ratio of correct colonies to the total number of checked colonies. Each experiment was performed in three biological replicates and data are presented as mean values ± standard error (SEM). **c** Summary of results for cloning uncharacterized BGCs using CAPTURE. BGCs ranging from 10 to 113 kb can be robustly cloned using the CAPTURE method at close to 100% efficiency regardless of their GC-content. Source data are provided as a Source Data file.

cells harboring helper plasmid pBE14. As shown in Fig. 1e, for the 50 kb linear DNA, the frequency of in vivo DNA circularization was ~33-fold higher than that of in vitro circularization and as the DNA size increased to 73 kb, this ratio grew to ~150-fold. These results indicate that due to creation of DNA by-products (i.e., DNA concatemers created in the ligation reaction), in vitro DNA circularization for long DNA molecules occurs at low frequencies. However, by eliminating formation of by-products, the circularization frequency can be significantly increased using an efficient in vivo DNA circularization system.

**Development of the CAPTURE method by characterizing various genomic DNA digestion/DNA assembly combinations**. To determine which genomic DNA digestion/DNA assembly combination will work most effectively with in vivo circularization for targeted cloning from microbial genomic DNA, we evaluated several systems. For genomic DNA digestion, we tested restriction enzymes and Cas12a. Unlike Cas9, Cas12a is able to create staggered cuts on DNA molecules[29] and the resulting cohesive ends can potentially be used for DNA assembly applications[30,31]. For DNA assembly, we examined three strategies: DNA ligation, isothermal Gibson assembly[19], and our newly developed DNA assembly approach termed T4 DNA polymerase exo + fill-in assembly (Fig. 2a). We determined the cloning efficiencies for four of the digestion and DNA assembly combinations (restriction enzymes/Gibson, *Fn*Cas12a/ligation, *Fn*Cas12a/Gibson, and *Fn*Cas12a/T4 exo + fill-in) using four high GC-content (72–77% range) BGCs from Actinomycetes with sizes ranging from 21-87 kb as model BGCs (Fig. 2b and Supplementary Table 1). For each cloning experiment, we selected at least seven colonies and analyzed the purified plasmids for each colony by restriction digestion. The cloning efficiencies were

calculated as the ratio of correct colonies to the total number of checked colonies. As shown in Fig. 2b, the *Fn*Cas12a/T4 exo + fill-in strategy exhibited close to 100% cloning efficiency for all four BGCs. Restriction enzymes/Gibson exhibited an average of ~63% cloning efficiency for 21, 38, and 62 kb target BGCs. However, we were unable to clone the 87 kb target BGC using this approach. *Fn*Cas12a/ligation exhibited close to 100% cloning efficiency for the 21 kb target BGC and ~57% efficiency for the 62 kb target BGC but only ~5% cloning efficiency for the 38 kb target BGC. We were also unable to clone the 87 kb target BGC using *Fn*Cas12a/ligation combination. The low efficiency of cloning for some target BGCs using *Fn*Cas12a/ligation may come from the fact that Cas12a enzymes do not generate defined sticky ends[30] and also possess non-specific trans-cleavage activity on dsDNA[32]. Consequently, ligation of sticky ends generated by Cas12a enzymes is not highly reliable. For *Fn*Cas12a/Gibson combination, we observed close to 100% cloning efficiency for the 21 kb target BGC and ~67% efficiency for the 38 kb target BGC. However, for cloning the larger target BGCs, this approach showed very low efficiencies (~10% for the 62 kb target BGC and ~5% for the 87 kb target BGC). Taken together, compared to the other three combinations, *Fn*Cas12a/T4 exo + fill-in, i.e., the CAPTURE method, exhibited the highest reliability and cloning efficiency for the four model BGCs.

**Large-scale cloning of uncharacterized BGCs and discovery of novel natural products**. To determine the overall robustness and cloning efficiency of CAPTURE, we used this method to clone a total of 43 uncharacterized natural product BGCs from 14 *Streptomyces* and three *Bacillus* species (Supplementary Figs. 7–13, Fig. 2c, and Table 1). The selected BGCs ranged from 10 to 113 kb in size. Out of the 43 selected target BGCs, we were able to

**Table 1 Summary of the results from the 43 uncharacterized BGCs cloned by CAPTURE.**

| Target BGC | Size | Predicted BGC type | Cloning efficiency | HPLC |
|---|---|---|---|---|
| *Streptomyces noursei* ATCC 11455 | | | | |
| C29 | 26 kb | Lanthipeptide | 100% | No new peak |
| C33 | 76 kb | NRPS | 100% | No new peak |
| *Streptomyces incarnatus* NRRL 8089 | | | | |
| C5 | 13 kb | Siderophore | 100% | No new peak |
| *Streptomyces leeuwenhoekii* NRRL B-24963 | | | | |
| C19 | 22 kb | Lassopeptide | 100% | No new peak |
| *Streptomyces griseochromogenes* ATCC 14511 | | | | |
| C6 | 59 kb | NRPS, type I PKS | 100% | 1 new peak |
| C7 | 113 kb | type I PKS, NRPS | 100% (4 colonies) | No new peak |
| C11 | 24 kb | Lassopeptide | 100% | No new peak |
| C15 | 11 kb | Siderophore | 100% | No new peak |
| C21 | 100 kb | NRPS, type I PKS, Lanthipeptide | 86% (12 colonies) | No new peak |
| C31 | 20 kb | Lassopeptide | 100% | No new peak |
| C39 | 103 kb | NRPS, type III PKS | 100% (10 colonies) | No new peak |
| C41 | 45 kb | type I PKS | 86% | No new peak |
| *Streptomyces griseoflavus* NRRL B-1830 | | | | |
| C28 | 65 kb | NRPS-like, type I PKS | 100% | No new peak |
| *Streptomyces* sp. NRRL B-11253 | | | | |
| C7 | 55 kb | Terpene-arylpolyene | 100% | No new peak |
| *Streptomyces* sp. NRRL F-5635 | | | | |
| C10 | 72 kb | type II PKS | 100% | No new peak |
| *Streptomyces* sp. NRRL F-6131 | | | | |
| C1 | 40 kb | type II PKS | 100% | 6 new peaks |
| C4 | 50 kb | NRPS | 100% | No new peak |
| C8 | 73 kb | type I PKS-NRPS | 100% | No new peak |
| C9 | 51 kb | NRPS | 100% | No new peak |
| C11 | 46 kb | type II PKS-NRPS | 100% | No new peak |
| C14 | 29 kb | Terpene | 100% | No new peak |
| C33 | 36 kb | type I PKS-NRPS | 100% | No new peak |
| *Streptomyces* sp. NRRL F-525 | | | | |
| C10 | 50 kb | T1PKS,hglE-KS | 57% | No new peak |
| C29 | 48 kb | NRPS-like | 100% | No new peak |
| *Streptomyces alni* NRRL B-24611 | | | | |
| C7 | 59 kb | type II PKS | 100% | No new peak |
| C11 | 46 kb | type II PKS | 100% | 4 new peaks |
| C21 | 65 kb | type I PKS-NRPS | 100% | No new peak |
| *Streptomyces bicolor* NRRL B-3897 | | | | |
| C6 | 47 kb | CDPS | 100% | No new peak |
| C15 | 67 kb | type II PKS | 100% | 3 new peaks |
| *Streptomyces cyaneofuscatus* NRRL B-2570 | | | | |
| C2 | 69 kb | type II PKS | 100% | 2 new peaks |
| C24 | 43 kb | NRPS-like | 100% | No new peak |
| *Streptomyces griseofuscus* NRRL B-5429 | | | | |
| C8 | 27 kb | type I PKS | 100% | No new peak |
| C11 | 46 kb | type I PKS | 100% | No new peak |
| C20 | 28 kb | type I PKS | 100% | No new peak |
| C21 | 35 kb | type I PKS | 71% | No new peak |
| C25 | 28 kb | type I PKS | 100% | 2 new peaks |
| C38 | 25 kb | type I PKS | 100% | 3 new peaks |
| *Streptomyces peuceticus* WC-3868 | | | | |
| C19 | 45 kb | type I PKS | 100% | No new peak |
| *Bacillus megaterium* DSM 319 | | | | |
| C2 | 26 kb | Phosphonate | 100% | No new peak |
| *Bacillus megaterium* QM B-1551 | | | | |
| C1 | 19 kb | Ladderane | 100% | No new peak |
| C2 | 19 kb | Lanthipeptide | 100% | No new peak |
| *Bacillus thuringiensis* YBT-020 | | | | |
| C1 | 14 kb | LAP | 100% | No new peak |
| C2 | 14 kb | Arylpolyene | 100% | No new peak |

successfully clone all BGCs without any failure. The CAPTURE method exhibited close to 100% cloning efficiency for all target BGCs tried including 11 target BGCs between 50 and 99 kb and three target BGCs larger than 100 kb (Table 1). These results indicate the CAPTURE method is an extremely robust and highly efficient direct cloning approach for cloning BGCs ranging from 10 to 113 kb in size regardless of their GC-content.

Next, we heterologously expressed all 43 cloned uncharacterized BGCs in either *Streptomyces avermitilis*, *Streptomyces lividans*, or *Bacillus subtilis*. HPLC analysis of crude extracts

from the 43 BGCs resulted in identification of seven BGCs with positive HPLC peaks (Fig. 3, Supplementary Figs. 14–18). To date, we have been able to identify the corresponding natural products for five of these seven BGCs. BGC #1 from *Streptomyces* sp. NRRL F-6131 encoding a putative type II polyketide synthases (PKSs) pathway was heterologously expressed in the model host *S. avermitilis* SUKA17. HPLC analysis of the metabolite profile led to the discovery of six cyclic head-to-tail heterodimers with a unique 5/6/6/6/5 pentacyclic carbon skeleton, designated as bipentaromycins A–F (**1–6**, Fig. 3a). The chemical structures of these compounds were elucidated by high-resolution electrospray ionization mass spectroscopy (HRE-SIMS) and nuclear magnetic resonance (NMR) (Supplementary Figs. 19–60, Supplementary Tables 2–4). One of these compounds was named as bipentaromycin E (**5**) and its molecular formula was established as $C_{45}H_{30}O_{13}$ by the negative HRESIMS ($m/z$ 777.1619 $[M - H]^-$, calcd for 777.1614, Supplementary Fig. 47), indicating 31 degrees of unsaturation. Detailed analyses of the $^1$H, $^{13}$C, and 2D NMR spectroscopic data revealed that **5** was a heterodimer and allowed the establishment and connectivity of units A (rings A/B/C/D/E) and B (rings A′/B′/C′/D′/E′) via C-2–C-8′ and C-8–C-2′. This assignment was further validated by X-ray crystallographic analysis with Flack parameter 0.007(5) (CCDC no. 1993850; Fig. 3a, Supplementary Table 5). We determined the antimicrobial activity of compounds **3–6** using a series of Gram-positive and Gram-negative bacteria including six ESKAPE pathogens commonly associated with antimicrobial resistance and bioweapon *Bacillus anthracis* as well as five commonly used fungal species (Supplementary Table 6). Interestingly, these compounds showed potent antibacterial activity towards both Gram-positive bacteria (e.g., *Bacillus anthracis* str. Sterne with minimal inhibitory concentration (MIC) of 4 µg/mL, methicillin-resistant *Staphylococcus aureus* USA300 with MIC of 32 µg/mL, vancomycinresistant *Enterococcus faecium U503* with MIC of 32 µg/mL) and Gram-negative bacteria (e.g., *Pseudomonas fluorescens* Pf-5 with MIC of 16–32 µg/mL). These four compounds showed different MIC values toward some of the bacteria (e.g., for *Staphylococcus epidermidis* 15 × 154, **3** and **4** had a MIC value of 4 µg/mL while **5** and **6** had a MIC value of 32 µg/mL). Notably, none of these compounds showed any antifungal activity. Metabolic profiling analyses of the other positive BGCs led to the isolation of three angucyclines (**7**, **8**, and **10**) with post-PKS tailoring enzyme-modified angular tetracyclic structural skeletons and an unexpected 2,3-*seco* tricyclic derivative (**9**), two anthraquinone or naphthoquinone derivatives (**11** and **12**) containing an acetylcysteine moiety, two known citreodiols and a oxidative derivative (**13–15**), and, to the best of our knowledge, novel molecules allenomycins A and B (**16** and **17**)[33] (Fig. 3b,c, Supplementary Figs. 61–106, Supplementary Tables 7–10).

To gain insight into the potential reasons for lack of compound production from 36 of the 43 cloned uncharacterized BGCs, we randomly selected 10 BGCs for transcriptional analysis in the heterologous host. For each of the 10 BGCs, we chose three targets corresponding to predicted core biosynthetic genes and performed RT-PCR analysis after 24 and 48 h of cell growth (Supplementary Figs. 107–111). For 6 of the 10 analyzed BGCs, we observed either very low or no detectable levels of RNA transcription after 30 cycles of RT-PCR for one or more analyzed biosynthetic genes. These results indicate that at least for 60% of the analyzed BGCs, lack or very low levels of transcription for the main biosynthetic genes plays an important role in lack of detectable compound production. As a result, strategies to increase levels of transcription such as insertion of strong promoters in front of core biosynthetic genes[34], overexpression of pathway-specific activators[35], or deletion of pathway-specific repressors[36] can potentially lead to

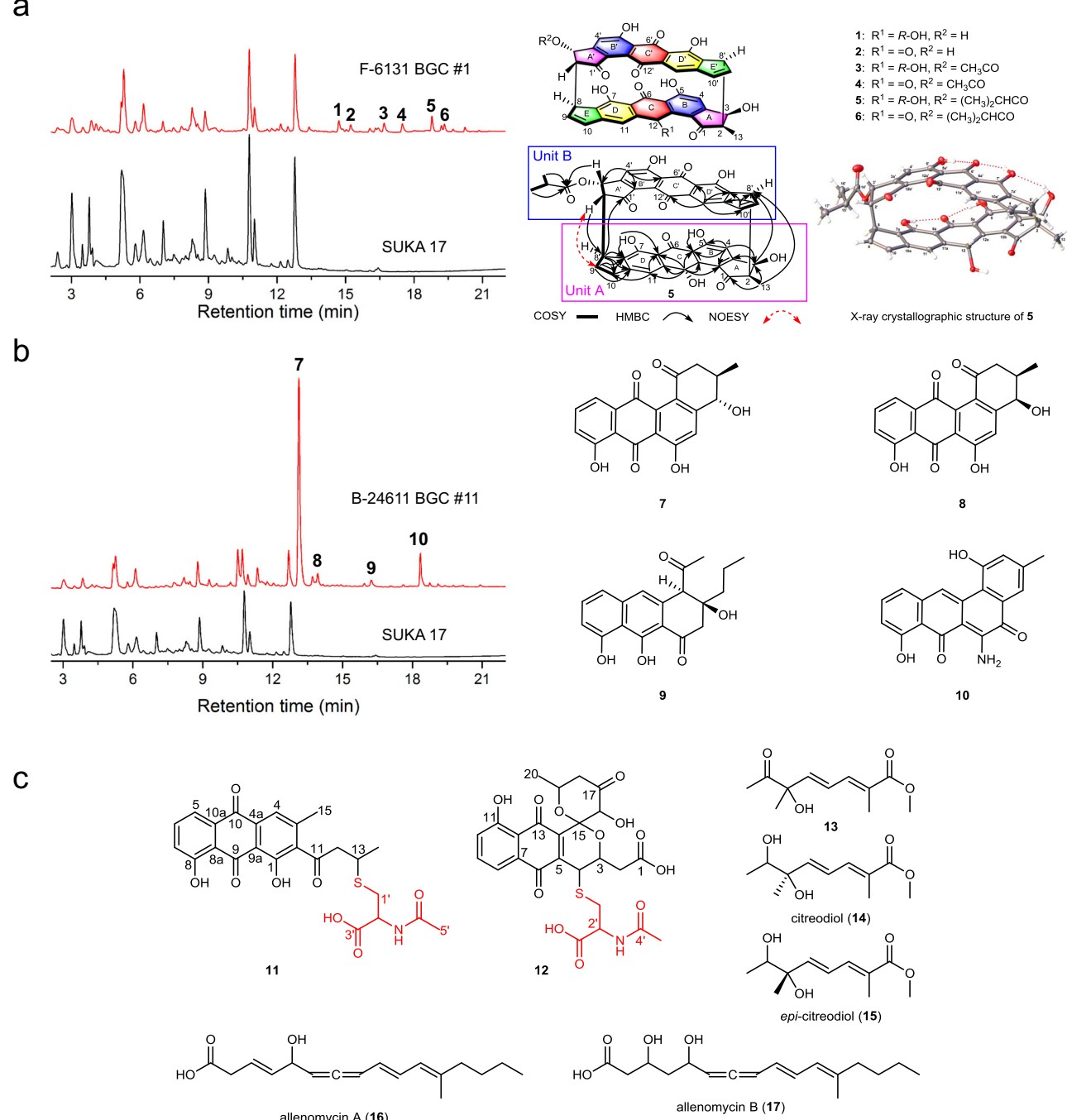

**Fig. 3 Metabolic profile and chemical structures of the characterized compounds from heterologous expression of five of the seven positive uncharacterized BGCs. a** HPLC analysis of the crude extract from heterologous expression of BGC #1 from *Streptomyces* sp. NRRL F-6131 in *S. avermitilis* SUKA17 and the chemical structures for the 6 heterodimers (**1–6**). **b** HPLC analysis of the crude extract from heterologous expression of BGC #11 from *Streptomyces alni* NRRL B-24611 in *S. avermitilis* SUKA17 and the chemical structure for the 4 new angucyclines and derivatives (**7–10**). **c** Chemical structures of the rest of characterized compounds including two new anthraquinone or naphthoquinone derivatives (**11** and **12**) containing an acetylcysteine moiety identified from *Streptomyces cyaneofuscatus* NRRL B-2570 BGC #2, two molecules allenomycin A and allenomycin B (**16** and **17**) identified from *Streptomyces griseofuscus* NRRL B-5429 BGC #25, two known citreodiols (**14** and **15**) and a new derivative (**13**) identified from *S. griseofuscus* NRRL B-5429 BGC #38.

the production of associated compounds from these BGCs in a heterologous host. For the remaining 40% of the BGCs, while low levels of transcription from other essential biosynthetic genes might still play a role in lack of compound production, a variety of previously described heterologous expression limiting factors such as absence of required precursors[37] or other missing host factors[14,38] can also explain lack of detectable compound production in the heterologous host.

## Discussion

By combining Cas12a digestion, T4 polymerase exo + fill-in DNA assembly, and in vivo Cre-*lox* DNA circularization, we were able to create an extremely robust and highly efficient method named CAPTURE for direct cloning of natural product BGCs. CAPTURE uses Cre-*lox* site-specific recombination instead of homologous recombination for the creation of circular DNA molecules which allows cloning targets with repetitive DNA

sequences. Moreover, the use of T4 polymerase exo + fill-in DNA assembly allows efficient assembly of DNA molecules with high GC-content. As a result, using CAPTURE, BGCs ranging from 10-113 kb regardless of their GC-content or repetitive DNA sequence can be directly cloned into vectors ready for heterologous expression in as short as 3-4 days. Compared to other direct cloning methods (see Supplementary Table 11), CAPTURE also offers high-levels of robustness and efficiencies. The high success rate of CAPTURE combined with short time-frame and low cost, makes this method suitable for large-scale cloning of microbial BGCs and discovery of novel natural products.

With the increasing number of sequenced microbial genomes, more and more uncharacterized natural product BGCs are being uncovered. However, identification and structure elucidation of the compounds associated with these BGCs has remained a major challenge. Direct cloning and heterologous expression represent the most efficient strategy for study of these uncharacterized BGCs and discovery of their associated compounds. As demonstrated in this study, direct cloning and heterologous expression of intact uncharacterized BGCs results in the production of associated compounds for ~16% of the BGCs. It was estimated that the genus of Streptomyces alone has the capacity to produce as many as 100,000 natural products with only a small fraction of them discovered so far[39]. Hence even with ~16% success rate for heterologous expression, by utilizing the large-scale cloning capabilities of CAPTURE, a significant number of natural products can be discovered at an accelerated rate. Discovery of bipentaromycins with strong antibacterial activity toward methicillin-resistant Staphylococcus aureus, vancomycinresistant Enterococcus faecium, and bioweapon Bacillus anthracis only provides an example of the multitude of uncharacterized bioactive compounds present in microorganisms. Therefore, we envision CAPTURE to become the method of choice for direct cloning of uncharacterized BGCs and large-scale discovery of novel natural products with unique bioactivities for use in drug development and agricultural applications.

## Methods

**Bacterial strains and reagents**. E. coli strain NEB10β (New England Biolabs, MA) was used for all cloning experiments. E. coli strains WM6026 and ET12567/pUZ8002 were used as conjugation donors for Streptomyces heterologous hosts. S. avermitilis SUKA17 and S. lividans TK24 were used for heterologous expression of natural product BGCs cloned from Actinomycete strains. B. Subtilis JH642 + sfp was used for heterologous expression of natural product BGCs cloned from Bacillus strains. E. coli conjugation donor WM6026 was a gift from William Metcalf (University of Illinois at Urbana-Champaign). S. avermitilis SUKA17 was a gift from Haruo Ikeda (Kitasato University, Japan). S. griseochromogenes ATCC 14511 was ordered directly from ATCC. All remaining Actinomycete strains were obtained from the Agricultural Research Service (NRRL) culture collection, Peoria, IL. All Bacillus strains were obtained from Bacillus Genetic Stock Center (Columbus, OH). Restriction enzymes, NEBuffers, T4 DNA polymerase, E. coli DNA ligase, dNTPs, NAD+, NEBuilder HiFi DNA assembly mastermix, and Q5 DNA polymerase were purchased from New England Biolabs (Ipswich, MA). All DNA oligonucleotides were ordered from Integrated DNA Technologies (Coralville, IA).

**Genomic DNA isolation**. Actinomycete strains were grown at 30 °C in 100 mL of modified MYG medium (10 g/L malt extract broth, 4 g/L yeast extract, 4 g/L glucose) using 250 mL baffled flasks containing glass beads. Bacillus strains were grown at 30 °C in 100 mL of LB medium using 250 mL baffled flasks. The cultures were allowed to reach late exponential—stationary growth phase and then centrifuged at $3220 \times g$ for 15 min. After removing the supernatant, the cells were fully resuspended in 12 mL of cell resuspension buffer (50 mM Tris-HCl pH 8.0, 25 mM EDTA) and transferred into 50 mL conical tubes. A combination of cell lysis enzymes (lysozyme 30 mg, achromopeptidase 20 mg (if necessary), mutanolysin 300 U (if necessary)), and 0.6 mg RNase A) was added to the sample and gently mixed. The mixture was incubated at 37 °C for 1-2 h to allow cell lysis and then 6 mg of proteinase K was added, gently mixed and the sample was continued to be incubated at 37 °C for 1 more hour. Following this incubation, 1.2 mL of 10% sodium dodecyl sulphate (SDS) was added, gently mixed and the sample was incubated at 50 °C for 1-2 h until the solution became clear. Recovery of genomic

DNA from cell lysate was performed by phenol-chloroform extraction. A total of 15 mL of phenol-chloroform-isoamyl alcohol (25:24:1, v/v, pH 8.0) was added to the cell lysate. The mixture was then placed on a rocker to allow gentle mixing without creating emulsion. Once the aqueous phase became completely white, the sample was centrifuged at $22,000 \times g$ for 45 min at room temperature. Without disturbing the precipitated proteins, the aqueous phase was transferred into a new 50 mL conical tube and 10 mL of chloroform was added. The sample was mixed as described above and centrifuged at $22,000 \times g$ for 10 min. Without disturbing the precipitated proteins, the aqueous phase was aliquoted into 1.7 mL centrifuge tubes and the DNA was recovered using isopropanol and sodium acetate precipitation. The DNA pellets were washed 3 times with 70% ethanol (v/v) and air dried. The DNA was rehydrated in 10 mM Tris-HCl pH 8.0 and incubated at 50 °C with occasional tube inversion until fully dissolved and stored at 4 °C.

**Expression and purification of FnCas12a enzyme**. The FnCas12a expression plasmid pET28-FnCas12a-TEV was obtained from Jin Wang (Shanghai Institute of Plant Physiology & Ecology, Chinese Academy of Sciences, China) and transformed into E. coli KRX (Promega, WI) according to manufacturer's protocol. The strain was cultivated overnight at 37 °C in 5 mL LB medium supplemented with 50 μg/mL kanamycin. Following overnight incubation, 2 mL of the overnight culture was added to 400 mL Terrific Broth containing 50 μg/mL kanamycin and incubated at 37 °C until the $OD_{600}$ of 1.2–1.5 was reached. The culture was then cold shocked by incubation in an ice bath for 15 min and protein expression was induced by addition of isopropyl β-D-1-thiogalactopyranoside (IPTG) and L-rhamnose to final concentrations of 1 mM and 0.1% (w/v), respectively. Expression was continued by incubation at 30 °C for 16-20 h. Cells were harvested by centrifugation at $5,000 \times g$ for 10 min and after removing the supernatant, the cell paste was stored at -80 °C until purification. For the purification step, frozen cells were thawed at room temperature and then fully resuspended in 30 mL of binding buffer (20 mM Tris-HCl pH 8.0, 1 M NaCl, 20 mM imidazole). The resuspended cells were lysed by sonication on ice for 10 min pulse time (30% amplitude, 5 s on, 5 s off). The solution was centrifuged 3 times at $20,000 \times g$ at 4 °C and the supernatant was used for purification using fast protein liquid chromatography (FPLC) and 1 mL Histrap HP column (GE Healthcare, IL). After equilibration with 5 column volume (CV) of binding buffer, the column was loaded with the supernatant and washed with 10 CV of binding buffer. The column was eluted with a gradient of 20 to 500 mM imidazole into 15, 1 mL fractions, and fractions containing protein of the expected size were pooled and concentrated with Amicon Ultra-15 50 kDA filtration units (MilliporeSigma, MA). The purified proteins were diluted to ~10 μM concentration using storage buffer (20 mM Tris-HCl pH 8.0, 150 mM NaCl, 15% (v/v) glycerol) and the aliquots were stored at –80 °C.

**DNA design for FnCas12a guide RNA and DNA receivers**. FnCas12a guide RNA molecules were transcribed from dsDNA templates with a size of 60 bp generated by hybridization of two ssDNA oligonucleotides. The dsDNA templates carry sequences for T7 promoter, FnCas12a gRNA scaffold, and a BGC specific spacer. For all guide RNA molecules, the spacer length of 18 nt was used and the spacer GC-content was kept in 20–72% range. In the selection of FnCas12a targets, to protect the generated BGC fragment ends after digestion from Cas12a's nonspecific trans-cleavage activity[32], the PAM sequence was kept inside the BGC fragment. The chosen gRNAs used for each BGC were checked by Basic Local Alignment Search Tool (BLAST) to avoid occurrence of undesired cleavage in the middle of the BGC fragment.

All DNA receivers were amplified using Q5 DNA polymerase following manufacturer's protocol using 1 ng of plasmid DNA template. For DNA receivers used in cloning target BGCs of larger than 90 kb, to avoid occurrence of false positive colonies, after PCR, 1 μL of DpnI restriction enzyme was added to the mixture and incubated at 37 °C for 45 min. The PCR products were run on 1% (w/v) agarose gel for 30 min at 120 V and purified by Zymoclean Gel DNA Recovery Kit (Zymo Research, CA) following manufacturer's protocol. For T4 DNA polymerase exo + fill-in assembly, each DNA receiver carried 39 bp of homology to the BGC fragment. This 39 bp homology included 24 bp sequence downstream of the PAM sequence (including the 18 bp of spacer), 3 bp of the TTV (V = A, C, or G) PAM sequence, and 12 bp sequence upstream of the PAM sequence. The GC-content for the 39 bp sequence was kept in 20–72% range while avoiding 4 or more consecutive G nucleotides. Examples of ssDNA templates and primers used for receiver amplification can be found in Supplementary Data 1.

**Guide RNA transcription and genomic DNA digestion**. For each guide RNA, 1 μL of each 100 μM forward and reverse template ssDNA oligonucleotides were mixed in a 20 μL reaction containing 2 μL of 10× NEBuffer 3.1. The mixture was incubated at 98 °C for 5 min and the temperature was then slowly lowered with the rate of 0.1 °C/s until 10 °C was reached. A total of 2 μL of the hybridized mixture was used as template for in vitro RNA transcription using HiScribe T7 quick high yield RNA Synthesis kit (New England Biolabs, MA) following manufacturer's protocol for transcription of short RNA molecules. After DNase treatment, the transcribed RNA was purified using RNA clean and concentrator kit (Zymo Research, CA) following manufacturer's protocol. The solution was diluted to ~800 ng/μL concentration and the aliquots were stored at −20 °C until digestion.

*Fn*Cas12a digestion of genomic DNA was carried out in a 300 μL reaction containing 15 μg purified genomic DNA, 2100 ng of each guide RNA, 60 pmol *Fn*Cas12a, and 30 μl of 10× NEBuffer 3.1. After mixing the components by gentle inversion, the reaction mixture was incubated at 37 °C for 2 h followed by 65 °C for 30 min. Then, 3 μL of 10 mg/mL RNase A was added, gently mixed and the mixture was incubated at 37 °C for 30 min. Following RNase treatment, the mixture was treated by 3 μL of 20 mg/mL proteinase K solution and incubated at 50 °C for 30 min. The reaction mixture was then transferred into 5PRIME PLG light phase-lock gel tubes (Quantbio, MA) and 300 μL of phenol-chloroform-isoamyl alcohol (25:24:1, v/v, pH 8.0) was added to the sample. The components were mixed by gentle tube inversion without creating emulsion. After visible precipitation of proteins in the aqueous phase, the sample was centrifuged at $20,000 \times g$ for 30 s. The aqueous phase was gently transferred into a new phase-lock tube and the extraction step was repeated with 300 μL of pure chloroform. The DNA was recovered using ethanol and sodium acetate precipitation and the DNA pellets were washed 2 times with 70% ethanol (v/v) and air dried. The DNA was rehydrated in 15 μL of 10 mM Tris-HCl pH 8.0 and incubated at 50 °C until fully dissolved.

Digestion of genomic DNA with restriction enzymes was carried out in a 300 μL reaction containing 15 μg purified genomic DNA, 60 U of each restriction enzyme, and 30 μL of appropriate 10x NEBuffer. The reaction mixture was incubated at the enzyme's optimal temperature for at least 4 h. After digestion, the mixture was incubated at 65 °C for 30 min and the DNA was recovered by phenol-chloroform extraction and ethanol precipitation as mentioned above.

**Preparation of electrocompetent *E. coli* cells for Cre-*lox* in vivo recombination.** *E. coli* NEB10β cells harboring pBE14 circularization helper plasmid were grown overnight at 30 °C in modified SOB medium (20 g/L Bacto tryptone, 5 g/L Bacto yeast extract, 10 mM NaCl, 2.5 mM KCl) supplemented with 8 μg/mL tetracycline hydrochloride. Following overnight incubation, 100 μL of the overnight culture was added to 10 mL of modified SOB medium supplemented with 8 μg/mL tetracycline hydrochloride and incubated at 30 °C. After 2 h of incubation (OD_{600} of ~0.2), 100 μL of 1 M L-arabinose was added to the culture to induce *gam* expression. The cells continued to grow at 30 °C until the OD_{600} of 0.45-0.55 was reached (~1.5 h after induction). The culture was then centrifuged at $3,220 \times g$ for 7 min at room temperature and the supernatant was immediately removed. The cells were gently resuspended in 1 mL of room temperature 10% (v/v) glycerol and centrifuged at $20,000 \times g$ for 30 s. Afterward, without disturbing the cell pellet, the supernatant was gently removed and this washing step was repeated 2 more times. The cells were then resuspended in a final volume of 70 μL 10% (v/v) glycerol using wide-bore pipette tips and transformed with DNA using Gene Pulser XCell Electroporation system (Bio-Rad, CA). For cloning BGCs, electroporation was performed using 1 mm cuvettes and 1250 V, 100 Ω, 25 μF condition[40].

For cloning target BGCs of larger than 50 kb, to increase DNA transformation efficiency, instead of modified SOB medium containing 5 g/L yeast extract, the overnight culture was added to modified SOB medium containing 1 g/L yeast extract. The induction and harvesting steps were carried out at the same OD_{600} as mentioned above.

**In vitro assembly of digested genomic DNA and DNA receivers.** Gibson isothermal assembly of DNA molecules was performed by adding 3-3.75 μg of digested genomic DNA, 15 ng of DNA receiver amplified from plasmid pBE44, 35 ng of DNA receiver amplified from plasmid pBE45, and 7.5 μL of 2× NEBuilder HiFi DNA assembly mastermix in a 15 μL reaction. After very gentle mixing using wide-bore pipette tips, the mixture was incubated at 50 °C for 90 min and stored at 10 °C until transformation.

For assembly of *Fn*Cas12a digested linear fragment using ligation, 500 ng of DNA receivers amplified from plasmids pBE44 and pBE45 were mixed with equimolar ratio in a 50 μL reaction containing 10 pmol *Fn*Cas12a, 350 ng of each guide RNA, and 5 μL of 10x NEBuffer 3.1. After mixing, the mixture was incubated at 37 °C for 1 h followed by 65 °C for 30 min. After RNase A and proteinase K treatment with concentrations mentioned before, digested fragments were purified using QIAquick PCR purification kit (Qiagen, Germany). DNA ligation was performed by gentle mixing of 3–3.75 μg of digested genomic DNA, 50 ng of purified digested receivers, 1.5 μL of *E. coli* DNA ligase buffer, and 0.75 μL of *E. coli* DNA ligase in a 15 μL reaction. The mixture incubated at room temperature for 150 min followed by transformation.

Assembly of linear DNA fragments by the T4 DNA polymerase exo + fill-in method was performed in a 15 μL reaction containing 3–3.75 μg of digested genomic DNA, 15 ng of DNA receiver amplified from plasmid pBE44, 35 ng of DNA receiver amplified from plasmid pBE45, 1.5 μL of NEBuffer 2.1, and 0.75 U of T4 DNA polymerase. Before adding T4 DNA polymerase, without any mixing, the sample was first incubated at 65 °C for 10 min followed by 25 °C hold. Afterward, T4 DNA polymerase was added. After very gentle mixing using wide-bore pipette tips, the mixture was incubated at 25 °C for 1 h, 75 °C for 20 min, and 50 °C for 30 min followed by 10 °C hold. Next, 1 μL of 1 mM NAD^+, 0.4 μl of 10 mM dNTPs, 1 μL (3 U) of T4 DNA polymerase, and 1 μL of *E. coli* DNA ligase were added and after gentle mixing using wide-bore pipette tips, the mixture was incubated at 37 °C for 1 h, 75 °C for 20 min, and stored at 10 °C until transformation.

For cloning target BGCs of larger than 50 kb, instead of pBE44, the DNA receivers were amplified from pBE48 plasmid and 44 ng of DNA was used in all assemblies. For cloning target BGCs from *Bacillus*, receivers were amplified from plasmid pBE50. For the first receiver (pBE50-f1) and second receiver (pBE50-f2), 35 ng and 18 ng of DNA were used in all assemblies, respectively.

For DNA transformation, the assembly mixture was dialyzed against ddH2O using membrane filters (MilliporeSigma, MA, cat. no. VSWP02500) for 30 min at room temperature. 10 μL of dialyzed mixture was then added to 70 μL of electrocompetent cells, mixed gently using wide-bore pipette tips, and added to electroporation cuvette. For cloning target BGCs of larger than 100 kb, instead of 10 μL, the entire dialyzed mixture was used for transformation.

**Analysis of cloned BGCs.** Upon electroporation, 1 mL of LB medium supplemented with 5 mM MgCl2 (ref. [41]) was added to the cuvette and the suspended cells were transferred into 14 mL round-bottom Falcon tubes. The culture incubated at 37 °C for 75 min with shaking at 250 rpm. All the cells were plated on LB agar plates containing appropriate antibiotics (50 μg/mL apramycin or 100 μg/mL carbenicillin) with blue/white screening and incubated at 37 °C until colonies appeared. For target BGCs of larger than 90 kb, the cells were plated on LB agar plates containing 100 μg/mL apramycin supplemented with 0.2% (w/v) glucose. For analysis of correct clones from plasmids harboring the 15A origin of replication, at least 7 single white colonies were picked and grown at 37 °C in LB medium supplemented with appropriate antibiotics. The plasmid DNA was purified from the cultures using Qiaprep Spin Miniprep Kit (Qiagen, Germany) following manufacturer's protocol. The purified plasmids were then digested by appropriate restriction enzymes and the samples were analyzed by agarose gel electrophoresis. A full DNA map for the cloned BGC plasmid was created based on the BGC and receivers DNA sequence and the resulting restriction digestion patterns were compared to the correct digestion pattern predicted by the SnapGene software. For analysis of correct clones from plasmids harboring the BAC origin of replication, at least 7 single white colonies were picked and suspended in 10 μL of sterile ddH2O. 1 μL of suspended cells were then checked by colony PCR using Phire Plant Direct PCR kit (ThermoFisher Scientific, MA) following manufacturer's protocol. The primers used for colony PCR were designed to check the correct assembly in the pBE48-BGC junction. The universal colony PCR primer (pBE48-colony-F) binds close to the end of pBE48 receiver. The other colony PCR primer is designed to bind the end of the BGC fragment close to pBE48 junction. Following colony PCR, at least one correct colony suspension was then randomly picked and transferred into 100 mL of LB medium supplemented with 0.2% (w/v) glucose and appropriate antibiotics and grown at 37 °C. The BACmid DNA was purified from the cultures using Plasmid Midi kit (Qiagen, Germany) following manufacturer's protocol. After Plasmid-Safe DNase (Lucigen, WI) treatment of purified DNA in Cutsmart buffer (New England Biolabs, MA), the DNA was digested by addition of appropriate restriction enzymes to the same mixture and the samples were analyzed by agarose gel electrophoresis.

**Bioinformatics analysis and selection of uncharacterized BGCs.** AntiSMASH[42] was used to analyze BGCs from both Actinomycete and *Bacillus* strains. The genome sequence of each strain was submitted to the AntiSMASH server with relaxed detection strictness, KnownClusterBlast and SubClusterBlast functions. Putative uncharacterized BGCs were selected based on low homology to known BGCs in addition of their core biosynthetic genes predicted by the AntiSMASH software. In choosing digestion sites by *Fn*Cas12a, the BGC boundaries were relaxed and were kept as broad as possible to include all putative biosynthetic genes and cluster specific regulators.

**Heterologous expression, product extraction, and HPLC analysis.** The correct recombinant plasmids containing BGCs from Actinomycetes were transformed into *E. coli* WM6026 (supplemented with 2,6-diaminopimelic acid at the final concentration 40 μg/mL for growth) or *E. coli* ET12567/pUZ8002. The plasmids were then transferred into *S. avermitilis* SUKA17 and *S. lividans* TK24 heterologous hosts by intergeneric conjugation according to previously described procedures[43]. Following conjugation, exconjugants were randomly selected and inoculated into liquid MYM medium (Malt extract 10 g/L, Yeast extract 4 g/L, Maltose 4 g/L, pH 7.2–7.4) and incubated at 28 °C for 4–5 days. BGC plasmids cloned from *Bacillus* strains were introduced into *B. subtilis* JH642 + sfp heterologous host by natural competence transformation. After transformation, recombinant strains carrying correct integration at the *amyE* locus were cultivated in liquid LB medium supplemented with 100 μg/mL spectinomycin for 3 days at 30 °C. After growth, the fermentation cultures from both *Streptomyces* and *Bacillus* heterologous hosts were harvested and centrifuged. The mycelia and supernatant were then separated and the mycelia were extracted with equal volume of acetone. After removal of the acetone by evaporation, the extract of mycelia and the supernatant were then extracted with equal volume of ethyl acetate. The ethyl acetate fraction was evaporated to dryness under reduced pressure. The crude extract was subjected to normal phase silica gel (200–400 mesh) column chromatography eluted with a gradient solvent system of hexane/ethyl acetate (from 100:0 to 0:100, v/v). Subfractions were further successively purified by Sephadex LH-20 column chromatography and reversed-phase semi-preparative high

performance liquid chromatography (HPLC) to yield compounds. The semi-preparative HPLC was carried out on an Agilent HPLC using the following solvent system: solvent A (water supplemented with 0.1% trifluoroacetic acid) and B (acetonitrile supplemented with 0.1% trifluoroacetic acid).

**RT-PCR analysis of uncharacterized BGCs in the heterologous host.** For analysis of BGCs in *S. avermitilis* heterologous host, each strain was first inoculated into liquid MYM medium and incubated at 28 °C for 48 h. 500 µL of the initial culture was then added to 50 mL of fresh MYM medium and incubated at 28 °C. Samples for RT-PCR analysis were taken after 24 h and 48 h of growth and the cell pellets were stored at -80 °C until RNA extraction. For analysis of BGCs in *B. subtilis* heterologous host, each strain was first inoculated into liquid LB medium supplemented with 100 µg/mL spectinomycin and incubated at 30 °C for 24 h. 500 µL of the initial culture was then added to 50 mL of fresh LB medium supplemented with 100 µg/mL spectinomycin and incubated at 30 °C. Samples for RT-PCR analysis were taken after 24 h and 48 h of growth and the cell pellets were stored at −80 °C until RNA extraction. For RNA extraction, 100 µl of frozen cell pellets were first grinded using a mortar and pestle cooled with liquid nitrogen. Total RNA was then purified from the ground cells using PureLink RNA mini kit (ThermoFisher Scientific, MA) following manufacturer's protocol. 4 µg of total RNA was first incubated with ezDNase enzyme (ThermoFisher Scientific) to remove any contaminating genomic DNA and reverse transcription was then performed by SuperScript IV VILO mastermix (ThermoFisher Scientific). The resulting cDNA product was used as template for RT-PCR analysis of different biosynthetic genes using Q5 DNA polymerase. For each biosynthetic gene, a 100-500 bp target was selected and the PCR products were analyzed by agarose gel electrophoresis. The primers used for PCR can be found in Supplementary Data 1. For each primer set, the PCR annealing temperature was first optimized using genomic DNA as template. *hrd*B and *rps*E genes were used as internal transcription controls for *S. avermitilis* and *B. subtilis* respectively.

**In vitro vs. in vivo circularization for long linear DNA molecules.** To perform the in vitro vs. in vivo circularization, 10 µg of the 50 and 73 kb plasmids (Supplementary Fig. 6) were first linearized by 120 U of restriction enzymes in 300 µL reactions by incubation at restriction enzyme's optimal temperature for at least 6 h. Following digestion, the reaction mixture was treated with 3 µL of 20 mg/mL proteinase K and purified by phenol:chloroform extraction in 5PRIME PLG light phase-lock gel tubes as described previously. The DNA was recovered using ethanol and sodium acetate precipitation and the DNA pellets were washed 2 times with 70% ethanol (v/v) and air dried. The DNA was rehydrated in 50 µL of 10 mM Tris-HCl pH 8.0 and incubated at 50 °C until fully dissolved. For in vitro circularization, 100 ng of purified linear DNA was mixed with 1 µL of T4 DNA ligase (New England Biolabs, MA) and 2 µL of 10x T4 DNA ligase buffer in a 20 µL reaction. The mixture was incubated at 16 °C for 18 h and then dialyzed against ddH2O using membrane filters (MilliporeSigma, MA) for 30 min at room temperature. 10 µL of the dialyzed mixture was then transformed into electrocompetent *E. coli* NEB10β cells prepared with the same method as described previously. For in vivo circularization, 100 ng of purified linear DNA was mixed with 2 µL of 10x T4 DNA ligase buffer in a 20 µL mixture. After dialysis for 30 min at room temperature, 10 µL of the dialyzed mixture was transformed into electrocompetent *E. coli* NEB10β cells harboring pBE14 helper plasmid prepared as described previously. For control, 100 ng purified circular plasmid was mixed with 2 µL of T4 DNA ligase buffer in a 20 µL mixture. After dialysis for 30 min at room temperature, 10 µL of the dialyzed mixture was transformed into electrocompetent *E. coli* NEB10β cells prepared with the same method as described previously. Different dilutions of the transformed cells were plated on LB agar plates supplemented with 50 µg/mL of apramycin and transformation efficiencies were calculated accordingly. Circularization frequencies were calculated as following:

$$\text{Circularization frequency (\%)} = \frac{\text{in vitro or in vivo transformation efficiency}}{\text{control circular transformation efficiency}}.$$

**Structure elucidation.** NMR data were analyzed by MestReNova software. HRESIMS data were analyzed by MassLynx software. X-ray single crystals were visualized by Olex2. Bipentaromycin structures were drawn using ChemBioDraw.

Compound **5** was obtained as a red crystal. The molecular formula of **5** was established as $C_{45}H_{30}O_{13}$ by the negative HRESIMS ($m/z$ 777.1619 [M − H]−, calcd 777.1614, Supplementary Fig. 47), indicating 31 degrees of unsaturation. Analysis of the $^1$H, $^{13}$C, and HSQC NMR spectroscopic data (Supplementary Figs. 48–50, Supplementary Table 4) of **5** revealed the presence of three methyls (two doublet methyls at $\delta_H$ 1.49 (d, $J = 7.0$) and 1.50 (d, $J = 7.0$), and a singlet tertiary methyl at $\delta_H$ 1.75), fifteen methines (including eight sp² hybrid olefinic methines and three oxygenated sp³ hybrid methines), and twenty-seven quaternary carbons including twenty olefinic ones and six carbonyl ones.

Detailed analyses of 2D NMR ($^1$H-$^1$H COSY and HMBC, Supplementary Figs. 51 and 52) data suggested that **5** was a heterodimer and allowed the establishment and connectivity of units A (rings A/B/C/D/E) and B (rings A′/B′/C′/D′/E′). In the unit A, HMBC correlations from H-4 to C-5/C-5a/C-12b, from H-11 to C-6/C-6a/C-7/C-7a/C-12, from H-12 to C-5a/C-6a/C-11/C-11a/C-12a/C-12b, from 5-OH to C-4/C-5/C-5a,

and from 7-OH to C-6a/C-7/C-7a indicated the presence of a 1,8,10-trihydroxy-9-anthrone (rings B/C/D). The existence and attachment of a 4-hydroxy-5-methylcyclopent-2-en-1-one ring (A) was supported by the HMBC correlations from H3-13 to C-1/C-2/C-3, from H-3 to C-3a/C-13, and from H-4 to C-3. In addition, a cyclopentadiene ring (E) was found in the unit A and fused to ring D, which were confirmed by the COSY correlations of H-8/H-9/H-10 and the HMBC correlations from H-8 to C-7a/C-9/C-10/C-10a, from H-9 to C-7a/C-8/C-10/C-10a, from H-10 to C-7a/C-8/C-9/C-10a, and from H-11 to C-10. Moreover, careful analysis of 1D and 2D NMR data of the unit B showed that it was highly similar to those of unit A. The differences were that the singlet methyl presented in unit A was replaced by a proton ($\delta_H$ 3.71, H-2′) in unit B, the C-12 ($\delta_C$ 62.3) hydroxyl group in unit A was changed to a keto group ($\delta_C$ 180.2, C-12′) in unit B, and an additional isobutyryl group was found to be located at C-3′. These assignments were corroborated by the COSY correlations of H-2′/H-3′ and H3-16′/H-15′/H3-17′, and the HMBC correlations from H-11′ to C-12′ and from H-3′/H-15′/H3-16′/H3-17′ to C-14′.

The connectivity of the two units via C-2–C-8′ was demonstrated by the key HMBC correlations from H-3/H3-13 to C8′ and from H-8′ to C-1/C-2/C-3. The linkage of the two units through C-8–C-2′ was verified by the COSY correlation of H-8/H-2′, and the HMBC correlations from H-8 to C-1′/C-2′ and from H-2′/H-3′ to C-8. Taken the above deduced structural information together, the planar structure of **5** was established. In **5**, the NOESY correlation (Supplementary Fig. 53) of H-9 and H-2′ supported that both of them had the same orientation. Finally, the absolute configuration of **5** was confidently determined by the single crystal X-ray diffraction analysis with Flack parameter 0.007(5) (CCDC no. 1993850; Supplementary Table 5), and the configurations of the seven chiral centers were characterized as 2 R, 3 R, 8 S, 12 R, 2′R, 3′S, 8′R.

The molecular formula of compound **7** was determined to be $C_{19}H_{14}O_6$ by HRESIMS ($m/z$ 337.0719 [M − H]−, calcd 337.0718, Supplementary Fig. 61). $^1$H NMR and $^{13}$C NMR spectroscopic data (Supplementary Figs. 62 and 63, Supplementary Table 7) of **7** were similar to those of fujianmycin A[44], except for an additional hydroxyl group attached to C-6 in **7**. This assignment was further supported by HMBC correlations (Supplementary Fig. 66) from H-5 to C-4/C-6a/C-12b, from H-4 to C-4a/C-5/C-12b, and from 4-OH to C-4a. The NOESY correlation (Supplementary Fig. 67) between Me-13 and H-4 indicated that they were in the same orientation. Biogenetically, considering that the coupling constant value ($^3J_{H3-H4}$, 9.3 Hz) between H-3 and H-4 was almost identical to that of fujianmycin A ($^3J_{H3-H4}$, 9.5 Hz), the absolute configurations of the chiral centers of **7** were assigned as 3 R and 4 S. Consequently, the structure of **7** was determined and designated as 6-hydroxyfujianmycin A1.

6-hydroxyfujianmycin A2 (**8**) has a same molecular formula $C_{19}H_{14}O_6$ as that of **7**, as determined by HRESIMS ($m/z$ 337.0717 [M − H]−, calcd 337.0718, Supplementary Fig. 68). The NMR spectroscopic data (Supplementary Figs. 69–74, Supplementary Table 7) of **8** were strikingly similar to those of **7**. The coupling constant value ($^3J_{H3-H4}$, 2.9 Hz) between H-3 and H-4 indicated it was *cis*-configuration. Moreover, the chemical shift changes of those carbons surrounding C-4 suggested that **8** was a stereoisomer of **7** at C-4 position. Therefore, the absolute configurations of **8** were tentatively deduced as 3 R and 4 R.

Compound **9** was isolated as a yellow needle crystal. Its molecular formula was established as $C_{19}H_{20}O_5$ by the negative HRESIMS ($m/z$ 327.1239 [M − H]−, calcd 327.1238, Supplementary Fig. 75). Detailed analyses of the 1D and 2D NMR spectroscopic data (Supplementary Figs. 76–81, Supplementary Table 8) of compound **9** revealed that the structure of **9** was similar to those of prejadomycin[45]. The main differences were that two olefinic carbons at $\delta_C$ 127.1 (C-2) and $\delta_C$ 157.2 (C-3) in prejadomycin disappeared, while one more methyl at $\delta_C$ 32.1 (C-12) and methylene at $\delta_C$ 15.3 (C-14) appeared in **9**, and concurrent chemical shift changes of the surroundings also occurred. The COSY correlations of H2-13/H2-14/H3-15 and HMBC correlations from H-4 to C-2/C-3/C-11 and from Me-12 to C-4/C-11 further supported this assignment. The NOESY correlation between H-4 and H2-13 indicated that they were cofacial and assigned as α-oriented. Accordingly, the structure of compound **9** was determined and the compound was named as 2,3-*seco*-prejadomycin.

Compound **10** was assigned a molecular formula as $C_{19}H_{13}NO_4$ by HRESIMS ($m/z$ 318.0771 [M − H]−, calcd 318.0772, Supplementary Fig. 82), implying the presence of the nitrogen atom. 1D and 2D NMR spectroscopic data (Supplementary Figs. 83–88, Supplementary Table 8) of compound **10** were similar to those of pregilvocarcin M-*o*-quinone and pregilvocarcin V-*o*-quinone[46]. The amino group presented and located at C-6 was supported by the COSY correlation between $\delta_H$ 9.06 and $\delta_H$ 10.88 and the HMBC correlations from H-12 to C-6a/C-12b and from 6-NH2 to C-6a. Finally, the structure of compound **10** was determined and the compound was named as pregilvocarcin W.

The molecular formula of **11** was determined to be $C_{24}H_{23}NO_8S$ by HRESIMS ($m/z$ 484.1093 [M − H]−, calcd for 484.1072, Supplementary Fig. 89). The NMR spectroscopic data (Supplementary Figs. 90–94, Supplementary Table 9) of **11** revealed that it was structurally similar to those of mansoquinone[47]. The only difference was the presence of an *N*-acetylcysteine moiety at C-13 in **11**. This assignment was supported by HMBC correlations from H-13 ($\delta_H$ 3.37) to C-1′ ($\delta_C$ 31.7), and from H2-1′ ($\delta_H$ 2.78/2.93) to C-13 ($\delta_C$ 35.0). Finally, the structure of **11** was determined and named as *N*-acetylcysteinmansoquinone.

The HRESIMS analysis gave a molecular formula of $C_{25}H_{25}NO_{12}S$ for **12** by HRESIMS ($m/z$ 562.1035 [M − H]−, calcd for 562.1025, Supplementary Fig. 95). The 1D and 2D NMR spectroscopic data (Supplementary Figs. 96–100,

Supplementary Table 9) were similar to those of griseusin B[48], except for the presence of an *N*-acetylcysteine moiety at C-4 and a ketone carbonyl group at C-17 in **12**. These assignments were confirmed by HMBC correlations from H-4 to C-1′, from $H_2$-1′ to C-4, as well as from H-16/$H_2$-18 to C-17 ($\delta_C$ 205.0). Finally, **12** was named as *N*-acetylcysteingriseusin.

Compound **13** was assigned a molecular formula of $C_{11}H_{16}O_4$ on the basis of HRESIMS ($m/z$ 195.1027 $[M − H_2O + H]^+$, calcd for 195.1016, Supplementary Fig. 101). The 1D and 2D NMR spectroscopic data (Supplementary Figs. 102–106, Supplementary Table 10) of **13** were highly similar to those of citreodiol. The only difference was that a hydroxyl group presented at C-7 ($\delta_C$ 73.1) in citreodiol was found to be changed to a keto group ($\delta_C$ 211.8) in **13**, which was further supported by HMBC correlations from $H_3$-8 ($\delta_H$ 2.22) and 6-Me ($\delta_H$ 1.42) to C-7. Ultimately, compound **13** was designated as 2,6-dimethyl-6-hydroxy-7-oxo-2,4-octadienoic acid methyl ester.

**Physiochemical properties for compounds 1–13**. Bipentaromycin A (**1**). Red powder; [1]H and [13]C NMR: see Supplementary Table 2; HRMS ($m/z$): (ESI/$[M − H]^-$) calcd. for $C_{41}H_{23}O_{12}$, 707.1195; found, 707.1219.

Bipentaromycin B (**2**). Red powder; [1]H and [13]C NMR: see Supplementary Table 2; HRMS ($m/z$): (ESI/$[M − H]^-$) calcd. for $C_{41}H_{21}O_{12}$, 705.1038; found, 705.1049.

Bipentaromycin C (**3**). Red powder; [1]H and [13]C NMR: see Supplementary Table 3; HRMS ($m/z$): (ESI/$[M − H]^-$) calcd. for $C_{43}H_{25}O_{13}$, 749.1301; found, 749.1320.

Bipentaromycin D (**4**). Red powder; [1]H and [13]C NMR: see Supplementary Table 3; HRMS ($m/z$): (ESI/$[M − H]^-$) calcd. for $C_{43}H_{23}O_{13}$, 747.1144; found, 747.1145.

Bipentaromycin E (**5**). Red crystal; [1]H and [13]C NMR: see Supplementary Table 4; HRMS ($m/z$): (ESI/$[M − H]^-$) calcd. for $C_{45}H_{29}O_{13}$, 777.1614; found, 777.1619.

Bipentaromycin F (**6**). Red powder; [1]H and [13]C NMR: see Supplementary Table 4; HRMS ($m/z$): (ESI/$[M − H]^-$) calcd. for $C_{45}H_{27}O_{13}$, 775.1457; found, 775.1464.

6-Hydroxyfujianmycin $A_1$ (**7**). Yellow powder; [1]H and [13]C NMR: see Supplementary Table 7; HRMS ($m/z$): (ESI/$[M − H]^-$) calcd. for $C_{19}H_{13}O_6$, 337.0718; found, 337.0719.

6-Hydroxyfujianmycin $A_2$ (**8**). Yellow powder; [1]H and [13]C NMR: see Supplementary Table 7; HRMS ($m/z$): (ESI/$[M − H]^-$) calcd. for $C_{19}H_{13}O_6$, 337.0718; found, 337.0717.

2,3-*Seco*-prejadomycin (**9**). Yellow crystal; [1]H and [13]C NMR: see Supplementary Table 8; HRMS ($m/z$): (ESI/$[M − H]^-$) calcd. for $C_{19}H_{19}O_5$, 327.1238; found, 327.1239.

Pregilvocarcin W (**10**). Yellow powder; [1]H and [13]C NMR: see Supplementary Table 8; HRMS ($m/z$): (ESI/$[M − H]^-$) calcd. for $C_{19}H_{12}NO_4$, 318.0772; found, 318.0771.

*N*-Acetylcysteinmansoquinone (**11**). Yellow powder; [1]H and [13]C NMR: see Supplementary Table 9; HRMS ($m/z$): (ESI/$[M − H]^-$) calcd. for $C_{24}H_{22}NO_8S$, 484.1072; found, 484.1093.

*N*-Acetylcysteingriseusin (**12**). Yellow powder; [1]H and [13]C NMR: see Supplementary Table 9; HRMS ($m/z$): (ESI/$[M − H]^-$) calcd. for $C_{25}H_{24}NO_{12}S$, 562.1025; found, 562.1035.

2,6-Dimethyl-6-hydroxy-7-oxo-2,4-octadienoic acid methyl ester (**13**). White powder; [1]H and [13]C NMR: see Supplementary Table 10; HRMS ($m/z$): (ESI/$[M − H_2O + H]^+$) calcd. for $C_{11}H_{15}O_3$, 195.1016; found, 195.1027.

**Antimicrobial assays**. The antimicrobial activities of bipentaromycins were measured against five fungi and 17 bacteria including nine pathogenic strains using broth microdilution method[49]. These indicator strains were *Saccharomyces cerevisiae* YSG50, *Saccharomyces cerevisiae* BY4741, *Aspergillus nidulans*, *Aspergillus niger*, *Aspergillus terreus*, *Bacillus halodurans* C-125, *Bacillus subtilis* ATCC 6633, *Escherichia coli* DH5a, *Lactococcus lactis* CNRZ 481, *Micrococcus luteus* ATCC 4698, *Pseudomonas fluorescens* Pf-5, *Pseudomonas putida* mt-2, *Streptococcus mutans* ATCC 25175, and pathogenic bacterial strains *Acinetobacter baumannii* ATCC 19606, *Bacillus anthracis* str. Sterne, *Bacillus cereus* TZ417, *Enterobacter cloacae*, vancomycin-resistant *Enterococcus faecium* U503 (VRE), *Klebsiella pneumoniae* ATCC 27736, *Pseudomonas aeruginosa* PA01, methicillin-resistant *Staphylococcus aureus* USA300 (MRSA), and *Staphylococcus epidermidis* 15×154. Fungi were inoculated from glycerol stocks and grown in 5 mL of YPAD on a rotary shaker (250 rpm) at 30 °C. *B. cereus*, *B. halodurans*, *B. subtilis* and *M. luteus* were grown in 5 mL of Tryptic Soy Broth (TSB) at 30 °C. *E. cloacae*, *P. fluorescens* and *P. putida* were grown in Nutrient Broth (NB) at 28–30 °C. *L. lactis* was grown in M17 Broth with 0.5% lactose at 30 °C. Other bacterial strains were grown in Brian Heart Infusion (BHI) at 37 °C. Overnight cultures were diluted into fresh sterilized medium and regrown to achieve an optical absorbance of 0.04 − 0.06 at 600 nm. The cells were diluted 10-fold in fresh medium to give the final concentration of ≈$10^6$ cfu $mL^{-1}$ and were distributed into 96-well microtiter plates, which were supplemented with compounds ranging from 128 to 0.25 µg/mL. The Minimum Inhibitory Concentrations (MIC) value was the concentration that resulted in no visible growth after 16–18 h cultivation. All experiments were tested in triplicates. Ampicillin (Amp) was used as a positive control for antibacterial assays.

**Reporting summary**. Further information on research design is available in the Nature Research Reporting Summary linked to this article.

## Data availability

Source data are provided with this paper. Deposition number of crystallographic data for **5** is CCDC 1993850. The data supporting the findings of this study are available within the article and its Supplementary Information files and all other data are available from the corresponding authors upon reasonable request. Source data are provided with this paper.

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

## Acknowledgements

We thank William Metcalf for helpful discussions and sharing plasmids and microbial strains. We are grateful to Furong Sun (School of Chemical Sciences Mass Spectrometry Laboratory) for HRESIMS data analysis, Xudong Guan (the Carl R. Woese Institute for Genomic Biology) for NMR measurement, and Toby J. Woods (School of Chemical Sciences Materials Chemistry Laboratory) for X-ray diffraction analysis. We thank Wenjun Zhang (South China Sea Institute of Oceanology, Chinese Academy of Sciences) and Haiyan Tian (Jinan University) for insightful discussion of structure elucidation. Some of this data was collected in the IGB Core on a 600 MHz NMR funded by NIH grant number S10-RR028833. This work was supported by grant GM077596 and AI144967 from the National Institutes of Health.

## Author contributions

B.E. and H.Z. conceived and designed the study. B.E., C.H. and H.Z. wrote the manuscript. B.E. performed the method development experiments. B.E., C.H., F.G., G.J, B.W. and S.K.T. performed the uncharacterized BGCs cloning experiments. C.H., F.G. and G.J. performed the heterologous expression and HPLC analysis experiments. C.H. performed structure elucidation of compounds and bipentaromycins antimicrobial activity assays. T.M. helped with heterologous expression experiments.

## Competing interests

The authors declare no competing interests.
