## [Peer Review File · Nature Communications]

Reviewers' Comments:

Reviewer #1:

Remarks to the Author:

This is an interesting and potentially important paper that describes an efficient method for harnessing orphan natural product biosynthetic gene clusters (BGCs). Efficiency of this Cas12a/Cre-lox recombination system is highly efficient. What continues to be a limiting factor is the number of BGCs that produce metabolites. Thus, in its current form this paper represents an incremental step forward, with only 16% of the cloned BGCs producing detectable secondary metabolites and only 4 of these resulting in isolation and characterization of the pure compounds. Among these, a new family of dimeric, head-to-tail pentacyclic ring system was fully characterized including a demonstration of moderate antibiotic activity. Although this is an important achievement, a significant number of questions remain unanswered. For example, of the 84% of BGCs that were transferred to a heterologous host and not expressed, do the authors have any insights into the reason for lack of metabolite expression? Was a BGC transcription analysis conducted on these strains? Is this a gene expression issue, precursor availability limitation, or some other problem? Of the three strains that showed metabolite expression but the molecules were not characterized, did the authors employ GNPS or other community metabolomics analysis to interrogate the extracts for known molecules or congeners of known metabolites? In terms of the bipentamycins, do the authors have insights into the basis for dimer formation, and can this question be readily probed to add depth to this study?

Reviewer #2:

Remarks to the Author:

The submitted manuscript reports a novel method for high-efficiency cloning of large natural product biosynthetic gene clusters. This approach uses a combination of Cas12a-based genomic DNA fragmentation, ligation of the targeted fragments with linear DNA receivers, and circularization of the resulting linear DNA molecules to form BGC-containing plasmids that can be used for heterologous expression of the BGCs. The developed approach was successfully adopted for cloning BGCs with a wide range of sizes. Based on these efforts, several new natural product BGCs were expressed in selected microbial hosts, which led to the identification of novel natural products with antimicrobial activities. Overall, this is a well-designed pioneering study with strong potential for future application in new natural product discovery. However, there are a few issues that need to be fully addressed before it can be accepted for publication.

Cloning efficiency is a critical parameter used in this work to compare the performance of the developed technique and other techniques. The definition of cloning efficiency thus needs to be explicitly stated in the main text to avoid confusion. For example, Fig. 2c shows that the cloning efficiency for 3 captured 100+ kb BGCs is 95%. Does it mean that there are at least 95% correct clones for each BGC, or 95% is the average efficiency taking into account all correct clones of 3 BGCs?

Along with comment above, the recombination frequency and circularization frequency in figure 1 also need to be clearly indicated.

Line 129-130. It is stated that "These results indicate that due to creation of DNA by-products, in vitro DNA circularization for long DNA molecules occurs at low frequencies" What are the specific by-products derived from the linearization treatment that affected in vitro DNA circularization? Please give a few specific examples here.

For identifying the best strategy for DNA digestion and assembly, four combinations were tested. To avoid the bias for the comparison, it would be helpful to include the combination of restriction enzymes/ligation, a classical cloning strategy, for this experiment.

In figure 1c. for the recombination efficiency using pBE12, the use of L-arabinose did not make significant difference. Since L-arabinose is the inducer for the gam gene expression, this result suggests that the leaky expression of pBAD promoter is sufficient for the recombination. Please discuss this in more details.

During the ssDNA hybridization at the T4 DNA polymerase exo + fill-in DNA assembly step, is it possible that there may be mismatches between the BGC fragment and the receivers? The clones generated via the mismatches may still produce the digestion pattern consistent with the correct clones, as the DNA sequence outside the hybridization regions is not affected. However, it may introduce undesired errors for BGC expression, e.g. when the errors are in the promoter region. Please elaborate on this potential issue. On the other hand, can these potential mismatches contribute to the cloning efficiency variation?

For the verification of the cloned BGCs, colony PCR was used. What did the colony PCR specifically check? A selected fragment of the BGCs, or a fragment of the receiver plasmid backbone? Please clarify this.

The confirmation of the correct clones relied on the restriction enzyme digestion analysis using DNA gels. what are the specific enzymes used for the restriction digestion analysis? The related information is not available in the main text or the supporting information. Also, what is the basis for evaluating whether the resulting DNA digestion pattern is correct? If such evaluation is done by analyzing the restriction sites of the BGC DNA sequences from gene databases, it should be indicated in the corresponding method section.

Also, for many incorrect clones, their digestion patterns are quite different from the correct ones. Can the authors explain what these incorrect clones are? Are they un-targeted fragments of the genomic DNA or something else?

For supplementary fig. 16, there is another new peak at close to 9 min. Did the authors analyze this peak?

For Antimicrobial assays described in the supplementary information, visible growth was used as the criterion for MIC measurement. This could be subjective and even lead to inaccurate results. Can the authors provide more details or evidence to justify the use of such an approach?

We would like to thank the reviewers for their time and constructive comments. Below are our point by point responses to the reviewers' comments.

Responses to Reviewer #1:

1. This is an interesting and potentially important paper that describes an efficient method for harnessing orphan natural product biosynthetic gene clusters (BGCs). Efficiency of this Cas12a/Cre-lox recombination system is highly efficient. What continues to be a limiting factor is the number of BGCs that produce metabolites. Thus, in its current form this paper represents an incremental step forward, with only 16% of the cloned BGCs producing detectable secondary metabolites and only 4 of these resulting in isolation and characterization of the pure compounds. Among these, a new family of dimeric, head-to-tail pentacyclic ring system was fully characterized including a demonstration of moderate antibiotic activity. Although this is an important achievement, a significant number of questions remain unanswered. For example, of the 84% of BGCs that were transferred to a heterologous host and not expressed, do the authors have any insights into the reason for lack of metabolite expression? Was a BGC transcription analysis conducted on these strains? Is this a gene expression issue, precursor availability limitation, or some other problem? Of the three strains that showed metabolite expression but the molecules were not characterized, did the authors employ GNPS or other community metabolomics analysis to interrogate the extracts for known molecules or congeners of known metabolites? In terms of the bipentarymycins, do the authors have insights into the basis for dimer formation, and can this question be readily probed to add depth to this study?

Response:

We would like to point out that the focus of this study was not on increasing the success rate of heterologous expression itself or study of the limitations of heterologous expression. The focus of this study was rather on increasing the speed and rate of natural product discovery through heterologous expression by developing a highly efficient and rapid direct cloning method.

As described in our results, heterologous expression of intact BGCs results in discovery of associated compounds from ~16% of the BGCs. While this number might appear to be low, when judging this number, all factors should be taken into account. The argument presented in this manuscript is that even if the success rate of heterologous expression is ~16%, if a large number of BGCs are cloned and heterologously expressed, this approach can lead to characterization of a plethora of novel natural products. Before the development of CAPTURE, the main bottleneck for studying a large number of BGCs was the lack of feasibility for cloning a large number of BGCs because of the low success rates or the amount of time and resources required. However, our development of CAPTURE now allows large scale cloning of natural product BGCs and solves that bottleneck. For example, if a microbial strain contains ~20 uncharacterized BGCs, all these BGCs can be cloned in ~10 days by CAPTURE which can result in discovery of associated compounds for ~3 BGCs in a relatively short period of time. *Streptomyces* species alone are suggested to have the capability to produce 100,000 natural products. Even if ~16,000 (~16%) of these natural products can be discovered by our approach,

it would still be a significant number. We modified the text in the discussion section to make this point clearer.

We understand that it would be ideal if heterologous expression of intact uncharacterized BGCs leads to production of their associated compounds for a large percentage of BGCs. However, a variety of factors which are currently unpredictable by only looking at the BGC sequence affect the success rate of heterologous expression and study of these factors is out of the scope of this study.

2. Of the 84% of BGCs that were transferred to a heterologous host and not expressed, do the authors have any insights into the reason for lack of metabolite expression? Was a BGC transcription analysis conducted on these strains? Is this a gene expression issue, precursor availability limitation, or some other problem?

Response:

We have not performed any analysis on the root cause for lack of compound production from these BGCs. As mentioned above, lack of compound production can be due to a variety of factors such as transcription, translation, and precursor or cofactor availability. As a result, we believe that the reason for lack of compound production for the 84% BGCs is most probably not universal and is different for each BGC and can include one or all of the mentioned causes. Since investigation of these factors is beyond the scope of this manuscript, we will report our findings in our follow-up manuscripts.

We would like to note that for heterologous expression, we used *Streptomyces avermitilis* SUKA17 for most of the experiments. This strain is capable of producing natural products with precursors derived from a variety of pathways including the sugar, polyketide, amino acid, and Shikimate pathways (Komatsu, Mamoru, et al. *ACS synthetic biology* 2.7 (2013): 384-396.). Use of this strain decreases the probability of precursor limitation. However, we would still be unable to rule out the possibility of precursor limitation for lack of compound production from some of the BGCs.

3. Of the three strains that showed metabolite expression but the molecules were not characterized, did the authors employ GNPS or other community metabolomics analysis to interrogate the extracts for known molecules or congeners of known metabolites?

Response:

We are currently in the process of characterizing the compounds produced by these three BGCs and we do not think GNPS or other community metabolomics analysis for the extracts is necessary at this stage. We have isolated several compounds from these BGCs but the structure elucidation work is not completed yet.

4. In terms of the bipentaromycins, do the authors have insights into the basis for dimer formation, and can this question be readily probed to add depth to this study?

Response:

As mentioned by the reviewer, the molecular basis for dimer formation of the bipentaromycins is fascinating. We are currently working on deciphering the biosynthetic mechanism for bipentaromyins and the results will be published in following studies.

Responses to Reviewer #2:

1. Cloning efficiency is a critical parameter used in this work to compare the performance of the developed technique and other techniques. The definition of cloning efficiency thus needs to be explicitly stated in the main text to avoid confusion. For example, Fig. 2c shows that the cloning efficiency for 3 captured 100+ kb BGCs is 95%. Does it mean that there are at least 95% correct clones for each BGC, or 95% is the average efficiency taking into account all correct clones of 3 BGCs?

Response:

To clear this confusion, we added additional text to the main text as well as Fig. 2 legend. The cloning efficiencies were calculated as the ratio of correct colonies to the total number of checked colonies (*e.g.* 7 correct colonies out of 7 checked colonies = 100% efficiency).

The 95% cloning efficiency in Fig. 2c describes the average cloning efficiency for all the three experiments (which is also stated in the figure). The specific cloning efficiency for each BGC can be found in Table 1. For the 100+ kb BGCs, all the checked colonies for two of the BGCs were correct. However, for one of the BGCs, 6/7 colonies were correct. As a result, in average, the three BGCs had ~95% cloning efficiency.

2. Along with comment above, the recombination frequency and circularization frequency in figure 1 also need to be clearly indicated.

Response:

The main text and Fig. 1 legend were changed to describe these frequencies more clearly. Recombination frequencies were calculated based on the ratio of white colonies to the total number of acquired colonies. Circularization frequencies were calculated based on the number of colonies acquired for each circularization experiment in comparison to the number of colonies acquired after transformation of the original circular DNA. The full description of this experiment can be found in the supplementary methods.

3. Line 129-130. It is stated that “These results indicate that due to creation of DNA by-products, *in vitro* DNA circularization for long DNA molecules occurs at low frequencies” What are the

specific by-products derived from the linearization treatment that affected in vitro DNA circularization? Please give a few specific examples here.

Response:

The by-products are not produced in the linearization treatment but rather in the circularization step with use of T4 DNA ligase. Similar to what is shown in Supplementary Fig. 2a, since the linearized plasmid created after digestion with restriction enzymes has compatible cohesive ends on its ends, incubation with T4 DNA ligase can either lead to circularization or creation of linear concatemers. As a result, the by-products referred to here are the concatemers created in the ligation step. The text was modified to explain this more clearly.

4. For identifying the best strategy for DNA digestion and assembly, four combinations were tested. To avoid the bias for the comparison, it would be helpful to include the combination of restriction enzymes/ligation, a classical cloning strategy, for this experiment.

Response:

The reason we did not test the restriction enzyme/ligation combination is that this approach is not going to work for this application. Generally speaking, due to their short recognition sequences, restriction enzymes cleave the genomic DNA in more than 100 locations. As the fragments generated by restriction enzymes all have the same sticky ends, the BGC fragment cannot be specifically selected by ligation. However, since Gibson assembly uses longer homology arms, it allows specific selection of the BGC fragment and that is why it was analyzed as a combination with restriction enzymes for direct cloning.

5. In figure 1c, for the recombination efficiency using pBE12, the use of L-arabinose did not make significant difference. Since L-arabinose is the inducer for the gam gene expression, this result suggests that the leaky expression of pBAD promoter is sufficient for the recombination. Please discuss this in more details.

Response:

We modified the main text to discuss this in more detail.

6. During the ssDNA hybridization at the T4 DNA polymerase exo + fill-in DNA assembly step, is it possible that there may be mismatches between the BGC fragment and the receivers? The clones generated via the mismatches may still produce the digestion pattern consistent with the correct clones, as the DNA sequence outside the hybridization regions is not affected. However, it may introduce undesired errors for BGC expression, e.g. when the errors are in the promoter region. Please elaborate on this potential issue. On the other hand, can these potential mismatches contribute to the cloning efficiency variation?

Response:

The homology arms for the receivers used in T4 DNA polymerase *exo* + fill-in DNA assembly are designed based on the exact sequence of BGC ends. As a result, we do not believe that there would be any mismatches in the ssDNA hybridization step. However, due to natural errors for polymerase enzymes, in rare cases, there might be errors in the fill-in step. Even if these errors occur, we do not believe that they have affected our experiments. Due to highly programmable nature of Cas enzymes, the cleavage location for the BGC can easily be changed to avoid cleavage at potentially important locations such as promoter regions for genes that are believed to be part of the BGC. As a result, while the probability of error is significantly low, its effect on BGC expression can easily be avoided in design of the guide RNA by the user and we made sure to clone extra sequences for each BGC to avoid this.

On the second note, we do not believe that the potential errors in the fill-in step would affect the cloning efficiency variation. Even if these errors occur, the digestion patterns would show correct cloning and only the homology arm sequence would show sequence errors.

7. For the verification of the cloned BGCs, colony PCR was used. What did the colony PCR specifically check? A selected fragment of the BGCs, or a fragment of the receiver plasmid backbone? Please clarify this.

Response:

The primers for colony PCRs were designed to check one of the junctions (pBE48-BGC) assembled by the T4 polymerase *exo* + fill-in. One of the primers binds to the end of pBE48 receiver while the other primer binds the end of BGC fragment close to pBE48 junction. The text in the methods section was modified to add this detail.

8. The confirmation of the correct clones relied on the restriction enzyme digestion analysis using DNA gels. What are the specific enzymes used for the restriction digestion analysis? The related information is not available in the main text or the supporting information. Also, what is the basis for evaluating whether the resulting DNA digestion pattern is correct? If such evaluation is done by analyzing the restriction sites of the BGC DNA sequences from gene databases, it should be indicated in the corresponding method section. Also, for many incorrect clones, their digestion patterns are quite different from the correct ones. Can the authors explain what these incorrect clones are? Are they un-targeted fragments of the genomic DNA or something else?

Response:

As requested, the restriction enzymes used for digestion analysis of each BGC were added next to the gel images in the Supplementary Info.

We generated a map of cloned BGC plasmid using the receivers and available BGC sequence by SnapGene software. We next compared the digestion patterns for purified plasmids with the correct digestion pattern predicted by the SnapGene software. The text in the methods section was modified to add this.

Almost all incorrect clones are empty receivers created by recombination of two linear receivers inside *E. coli* cells. Since the occurrence of other incorrect clones was very rare, we did not analyze how they were exactly created.

9. For supplementary fig. 16, there is another new peak at close to 9 min. Did the authors analyze this peak?

Response:

We are currently in the process of characterizing the compounds produced from this BGC. Although in this image, it might look like there is another peak close to 9 min, this peak is also seen in the *S. lividans* TK24 host itself but with different intensity. As a result, this peak is not actually a new peak.

10. For Antimicrobial assays described in the supplementary information, visible growth was used as the criterion for MIC measurement. This could be subjective and even lead to inaccurate results. Can the authors provide more details or evidence to justify the use of such an approach?

Response:

The protocol we used in this manuscript is a standard and commonly used protocol mentioned in the reference (Tietz, J. I *et al.* A new genome-mining tool redefines the lasso peptide biosynthetic landscape, *Nature Chemical Biology* **13**, 470-478 (2017)). Unfortunately, these compounds were not as significantly active as those commercial antimicrobial agents against those tested strains. As a result, we did not use a much more elaborate approach to read each plate.

The reference was added to the supplementary methods.

Reviewers' Comments:

Reviewer #1:

Remarks to the Author:

I thank the authors for their careful consideration of the critiques and the additional experimental work conducted to improve the manuscript.

Reviewer #2:

Remarks to the Author:

The revised manuscript has addressed the previous review comments and accordingly, new discussion and modifications were added to the manuscript. This manuscript in its current format provides new insights and an innovative cloning technique for promoting research involving natural product biosynthetic gene clusters. Due to its significance and importance to the related research communities, this revised manuscript is recommended for publication.